# A Simple, Improved Method for Scarless Genome Editing of Budding Yeast Using CRISPR-Cas9

**DOI:** 10.3390/mps5050079

**Published:** 2022-10-04

**Authors:** Rhiannon R. Aguilar, Zih-Jie Shen, Jessica K. Tyler

**Affiliations:** 1Department of Pathology and Laboratory Medicine, Weill Cornell Medicine, New York, NY 10065, USA; 2Weill Cornell/Rockefeller/Sloan-Kettering Tri-Institutional MD-PhD Program, New York, NY 10065, USA

**Keywords:** CRISPR-Cas9, budding yeast, homologous recombination, point mutation, genome editing, genetic engineering

## Abstract

Until recently, the favored method for making directed modifications to the budding yeast genome involved the introduction of a DNA template carrying the desired genetic changes along with a selectable marker, flanked by homology arms. This approach both limited the ability to make changes within genes due to disruption by the introduced selectable marker and prevented the use of that selectable marker for subsequent genomic manipulations. Following the discovery of CRISPR-Cas9-mediated genome editing, protocols were developed for modifying any DNA region of interest in a similar single transformation step without the need for a permanent selectable marker. This approach involves the generation of a DNA double-strand break (DSB) at the desired genomic location by the Cas9 nuclease, expressed on a plasmid which also expresses the guide RNA (gRNA) sequence directing the location of the DSB. The DSB is subsequently repaired via homologous recombination using a PCR-derived DNA repair template. Here, we describe in detail an improved method for incorporation of the gRNA-encoding DNA sequences into the Cas9 expression plasmid. Using Golden Gate cloning, annealed oligonucleotides bearing unique single-strand DNA overhangs are ligated into directional restriction enzyme sites. We describe the use of this CRISPR-Cas9 genome editing protocol to introduce multiple types of directed genetic changes into the yeast genome.

## 1. Introduction

The popularity of budding yeast (*Saccharomyces cerevisiae*) as a model organism for research is largely due to its genetic tractability, a direct consequence of having highly active homologous recombination (HR) machinery. This enables the incorporation of DNA sequence changes that originate from exogenous DNA templates bearing 40 bp or more homology to the flanking genomic sequences to facilitate integration via HR. This is termed one- or single-step gene engineering [1] and occurs without the need for generation of a genomic DNA double-strand break (DSB). However, the frequency of recombination events that result in the incorporation of a template into the genome is relatively low in yeast, occurring at 1 in 10^6^−10^7^ with 40 bp of flanking homology [2,3], requiring the use of a selectable marker to identify cells with the newly introduced DNA. There are several limitations to this approach. Firstly, a limited number of selectable markers exist for selecting for genomic insertion events [1,4,5,6]. As the selectable marker remains in the genome, this makes it challenging to generate yeast strains with multiple genetic changes, due to the limited number of available markers to select for each subsequent modification. While re-use of a specific selectable marker is possible, it requires a two-step protocol, usually involving recycling of an *URA3* marker cassette [7,8,9]. Secondly, while single-step genetic engineering is facile for genomic changes including yeast gene deletions, promoter changes, and for introducing sequences resulting in protein fusions, it leaves behind a selectable marker immediately adjacent to the engineered site. This is problematic for genetic engineering that requires the surrounding sequences not be disrupted by a selectable marker. One way around this is to use multiple rounds of PCR amplifications to generate a template containing both the desired mutation and a selectable marker either upstream or downstream of the sequences that cannot be disrupted [10]. For example, to generate a mutant protein, the template will need to be generated with either the selectable marker upstream of the promoter or downstream of the transcription termination sequences. In this way point mutations can be incorporated, but it is quite laborious [1,10]. As a result, many labs use a plasmid [9] to express a mutant allele in a strain where the endogenous gene is deleted, which can result in uneven gene expression as a result of variable plasmid copy number [1].

The development of CRISPR-Cas9 protocols has streamlined the generation of mutations at any region in the yeast genome. A guide RNA (gRNA) sequence, designed to target the desired region of the genome to be altered, directs Cas9 to make a DSB in the vicinity of the sequence homologous to the gRNA. To direct repair using HR machinery a linear PCR-generated dsDNA template, carrying both the desired mutation and homology to the native target locus, is co-transformed along with a plasmid encoding Cas9 and the gRNA [11,12]. The plasmid expressing both gRNA and Cas9 bears a selectable marker but does not integrate into the genome. Due to the fact that only 1 in 10^2^ yeast cells are successfully transformed per microgram of plasmid DNA [13], the selectable marker is used to ensure selection for only those yeast that have been transformed with the plasmid + template DNA. This has rendered multi-step protocols for selection marker recycling unnecessary, as the Cas9-encoding plasmid is diluted out by cell divisions after gene editing is complete, rendering the marker available for future use [14,15]. Furthermore, as the DSBs produced by Cas9 are poorly compatible with yeast cell survival, specific selection for the genetically altered yeast is not necessary as selection naturally occurs for those cells which have successfully performed homologous recombination after a Cas9-directed DSB [14,16,17]. 

While multiple methods of introducing Cas9 and gRNA into cells exist, including the direct transfection of a Cas9-gRNA ribonucleoprotein (RNP) into the nucleus of mammalian cells [18], such techniques have not been developed for budding yeast. For the direct transfection of proteins into yeast, the cell wall must first be digested [19]—a delicate and time-consuming step. The simplicity of DNA transformation, by contrast, makes the introduction of a plasmid containing Cas9 and the gRNA sequence the favored method. Several yeast CRISPR-Cas9 methods have been developed, and these methods noticeably differ in the steps used to clone a gRNA into the Cas9 expression vector. For example, gRNA exchange by restriction enzyme-free cloning relies on a set of complementary primers containing the new gRNA sequence and homology to the plasmid sequence both upstream and downstream, which are used to PCR-amplify the entire 8kb+ plasmid (termed the pCAS plasmid) [20]. Amplification of such a long DNA sequence raises the possibility of unwanted mutations being introduced into the genes encoding Cas9, the promoters, or the selection marker, that may compromise the accuracy and efficiency of genome editing. 

To avoid potential mutations resulting from PCR errors, other methods have been developed using restriction enzymes to clone the gRNA into the Cas9-expressing plasmid. A method based on traditional cloning with 2 restriction enzymes [15], while effective and simple to perform in most laboratories, is a multi-step process with many places for technical errors. Alternatively, the speed and simplicity of Golden Gate cloning using type II-S restriction enzymes that enable insertion of DNA fragments in the correct orientation using a single restriction enzyme [21] has been used to generate gRNA/Cas9 expression vectors. One such method involves a 3-step process toward generating the gRNA/Cas9-encoding plasmid, where each intermediate ligation must be transformed into *E. coli*, isolated from *E. coli* and validated, an unnecessarily time-consuming process [22]. 

Direct placement of type II-S restriction enzyme sites on either side of the gRNA sequence has been used to allow for a single Golden Gate cloning step to replace the gRNA with a new gRNA [23]. A previously published method, however, makes use of a galactose-inducible promoter (pGal1) for expression of Cas9. The ability to control timing of Cas9 expression in the yeast is not necessary in most circumstances, because the use of a constitutive promoter enables efficient DSB generation and efficient genome editing. Furthermore, driving Cas9 expression from the galactose inducible Gal1 promoter requires changing the media [24] and if used in a yeast strain that already contains an important gene under the control of the a Gal-inducible promoter, this may be undesirable.

Here, we present a gene editing method using a modification of the pCAS plasmid first published by Ryan et. al. [14,20] (pCAS was a gift from Jamie Cate (Addgene plasmid #60847; http://n2t.net/addgene:60847 (accessed on 25 September 2022); RRID:Addgene_60847)), a vector which constitutively expresses Cas9 and the gRNA. We have inserted a pair of BsaI sites flanking the gRNA insertion site, enabling the directional ligation of annealed oligonucleotides encoding the gRNA after digestion with BsaI. The resulting plasmid, pCASB (Figure 1A), allows for rapid and accurate exchange of the gRNA in less than an hour and eliminates the risk of PCR errors resulting from restriction enzyme-free cloning. Additionally, we present a detailed method for use of this plasmid, in conjunction with a PCR-derived DNA template, for multiple types of site-directed mutagenesis in a target *S. cerevisiae* gene.

## 2. Experimental Design

Type II-S restriction enzymes such as BsaI are ideal for Golden Gate cloning due to their ability to cut the DNA outside of their restriction sites, leaving behind unique overhangs that enable directional cloning. The pair of BsaI cut sites flanking the gRNA encoding locus in the pCASB plasmid thus create asymmetric overhangs after digestion with BsaI (Figure 1B), ensuring that the new gRNA will be ligated into the plasmid in the correct orientation. 

A schematic describing the steps involved when using pCASB for CRISPR-Cas9 site-directed mutagenesis is presented in Figure 1C. Briefly, the gRNA-encoding DNA is generated by annealing the appropriate oligonucleotides to generate a dsDNA with ssDNA overhangs complimentary to the BsaI digested pCASB plasmid (Figure 1B). The details of the oligonucleotide design are described in detail in Section 3.1 below. The ligation reactions are transformed into *E. coli* and plasmid DNA is isolated from Kanamycin-resistant clones. Following confirmation of the isolated plasmid with the desired gRNA encoding gene, the plasmid is transformed into yeast cells along with a linear homology directed repair (HDR) dsDNA template containing the desired genetic mutation (discussed below). 

The entire protocol takes approximately 2 weeks and can be completed in any laboratory set up for bacterial and yeast culture, subcloning, and PCR. Many portions of the method can be modified to suit the lab’s preferred methods, such as for yeast transformation and genomic DNA isolation.

### 2.1. Materials

The materials used in this protocol include:0.2 mL PCR tubes (USA Scientific, Ocala, FL, USA, Cat. no.: 1402-4708)1.7 mL microcentrifuge tubes (VWR, Radnor, PA, USA, Cat. no.: 87003-294)Budding yeast strain to be genetically engineered (must NOT already contain a kanamycin resistance marker, conferring resistance to G418 in yeast)pCASB Plasmid (Addgene ID: 190175)Oligonucleotides encoding desired gRNA (described in Section 3.1)Oligonucleotide annealing buffer, such as TE (described below) containing 50 mM sodium chloride (Fisher Scientific, Hampton, NH, USA, Cat. no.: S271)BsaI-HF^®^ v2 (New England Biolabs, Ipswich, MA, USA, Cat. no.: R3733)T4 DNA Ligase and buffer (New England Biolabs, Ipswich, MA, USA, Cat. no. M0202)Filtered water (such as Millipore), sterilized by means of autoclaving where indicatedHigh-efficiency competent *E. coli*, such as One Shot TOP10 Chemically Competent *E. coli* (Thermo Fisher Scientific, Waltham, MA, USA, Cat. no.: C404010)14 mL culture tubes for bacterial culture growth (Corning, Corning, NY, Cat. no.: 352051). Autoclaved to sterilize.Kanamycin Sulfate (Sigma-Aldrich, St. Louis, MO, USA, Cat. no.: 60615), prepared to 400 mg/mL (1000× stock solution) in sterile water0.2-micron syringe filters (VWR, Radnor, PA, USA, Cat. no.: 76479-024), 50 mL conical tube vacuum filters (MilliporeSigma, Burlington, MA, USA, Cat. no.: SCGP00525), and 250–500 mL flask vacuum filters (Corning, Corning, NY, USA, Cat. no.: 430196-7) for sterilization of reagentsLB liquid media
○LB Powder (IBI Scientific, Dubuque, IA, USA, Cat. no.: IB49020), 25 g/L in water. Autoclaved to sterilize.LB agar plates containing 400 μg/mL Kanamycin
○Petri dishes (Fisher Scientific, Hampton, NH, USA, Cat. no.: FB0875713)○Agar A (Bio Basic, Amherst, NY, USA, Cat. no.: FB0010)○Add 15 g/L Agar A to LB liquid media before autoclaving. Allow to cool to 55 °C before adding 1 mL/L of Kanamycin stock solution and pouring into Petri dishes.Glass beads or equivalent for plating (Zymo Research, Irvine, CA, Cat. no.: S1001). Autoclaved to sterilize.Plasmid miniprep kit (Zymo Research, Irvine, CA, USA, Cat. no.: D4016)Oligonucleotides or other DNA encoding template sequence for HDR (described in Section 3.3)High-fidelity DNA polymerase enzyme and buffer, such as Expand^™^ High Fidelity PCR System (EHIFI-RO Roche, Sigma Aldrich Cat. no.: 11732650001)10 mM dNTPs (Promega, Madison, WI, USA, Cat. no.: U1515)100% ethanol (VWR, Radnor, PA, USA, Cat. no.: 89125-186)70% ethanol (diluted in filtered water)Sodium Acetate (Thermo Fisher Scientific, Waltham, MA, USA, Cat. no. BP333-500), 3 M solution in waterTAE buffer (40 mM Tris-acetate, 1 mM EDTA)
○Tris Base (Thermo Fisher Scientific, Waltham, MA, USA, Cat. no.: BP152)○Glacial acetic acid (Sigma-Aldrich, St. Louis, MO, USA, Cat. no.: A6283)○For 1 L of 50× solution, combine 242 g Tris Base, 57.1 mL glacial acetic acid, and 100 mL 0.5 M EDTA, pH 8.0. Add water to 1 L)Quick dissolve agarose (Genesee Scientific, San Diego, CA, USA, Cat. no.: 20-102QD)
○To prepare DNA gel, mix 1% *w*/*v* agarose in TAE buffer, microwave to boil and dissolve agarose powder. Cool slightly before adding SYBR safe DNA gel stain (10,000×) and pouring into gel mold.SYBR™ Safe DNA Gel Stain (Thermo Fisher Scientific, Waltham, MA, USA, Cat. no.: S33102)DNA loading dye (New England Biolabs, Ipswich, MA, USA, Cat. no.: B7024)100 bp DNA ladder (New England Biolabs, Ipswich, MA, USA, Cat. no.: N3231)TE Buffer (10 mM Tris-HCl pH 8.0, 0.1 mM EDTA)
○Tris Base (Thermo Fisher Scientific, Waltham, MA, USA, Cat. no.: BP152)○EDTA (Caisson Laboratories, Smithfield, UT, USA, Cat. no.: E004)○For 500 mL of 100× solution, combine 60.55 g Tris Base with 1.86 g EDTA Na_2_ • 2H_2_O. Add filtered water to under 500 mL, adjust pH to 7.8 with concentrated HCl (Thermo Fisher Scientific, (Waltham, MA, USA, Cat. no.: A144), then fill to 500 mL with filtered water. Use a 0.2-micron filter to sterilize.Yeast transformation buffer: 1× TE, 100 mM Lithium Acetate
○Lithium acetate (Sigma-Aldrich, St. Louis, MO, USA, Cat. No.: 517992), 1 M stock solution in water○For 50 mL, mix 5 mL of 10× TE, 5 mL 1 M lithium acetate, and 40 mL of filtered water. Use a 0.2-micron filter to sterilize.Deoxyribonucleic acid, single stranded from salmon testes, 10 mg/mL (Sigma-Aldrich, St. Louis, MO, Cat. no.: D9156)PEG Solution: 40% Poly(ethylene) glycol in yeast transformation buffer
○Poly(ethylene) glycol (Sigma-Aldrich, St. Louis, MO, USA, Cat. no.: P4338)○For 50 mL, combine 25 g poly(ethylene) glycol, 5 mL 1× TE, and 5 mL 1 M lithium acetate. Add warm (50–60 °C) filtered water to 50 mL. Mix until dissolved. Use a 0.2-micron filter to sterilize.Dimethyl Sulfoxide (DMSO) (Sigma-Aldrich, St. Louis, MO, USA, Cat. no.: D8418)YPD liquid media
○20 g/L Peptone (Research Products International, Mount Prospect, IL, USA, Cat. no.: P20250)○10 g/L Yeast Extract (Research Products International, Mount Prospect, IL, USA, Cat. no.: Y20025)○20 g/L Dextrose (Fisher Scientific, Hampton, NH, USA, Cat. no.: D16-1)○Mix in filtered water, autoclave to sterilize.Glass tubes, 15 mL (Globe Scientific, Mahwah, NJ, USA, Cat. no.: 1517), and lids (DWK Life Sciences, Milville, NJ, Cat. no.: 73660-16) for yeast culture. Autoclave to sterilize.Geneticin (G418) sulfate (Santa Cruz Biotechnology, Dallas, TX, USA, Cat. no.: sc-29065), prepared to 200 mg/mL (1000× stock solution) in autoclaved filtered water.YPD Agar plates
○Add 15 g/L Agar A (Bio Basic, Amherst, NY, USA, Cat. no.: FB0010) to YPD liquid media before autoclaving.YPD Agar plates containing G418
○After autoclaving, allow liquid agar to cool to 55 °C before adding 1 mL/L of G418 stock solution.Wooden dowels (Puritan Medical Products, Guilford, ME, Cat. no.: 807), toothpicks, or pipette tips, for picking colonies. Autoclave to sterilize.Yeast genomic prep buffer (1% *w*/*v* SDS, 2% *v/v* Triton X-100, 100 mM NaCl, 10 mM Tris HCl pH 7.9, 1 mM EDTA)
○SDS (Promega, Madison, WI, USA, Cat. no.:H5113)○Triton X-100 (Sigma-Aldrich, St. Louis, MO, USA, Cat. no: X-100)○Sodium chloride (Fisher Scientific, Hampton, NH, USA, Cat. no.: S271)0.5 mm zirconia/silica beads (BioSpec Products, Bartlesville, OK, USA, Cat. No.: 11079105Z)Phenol:chloroform:isoamyl alcohol, 25:24:1 Saturated with 10 mM Tris, pH 8.0, 1 mM EDTA (Sigma-Aldrich, St. Louis, MO, USA, Cat. no.: P2069)Taq DNA polymerase (New England Biolabs, Ipswich, MA, USA, Cat. no.: M0273), or similar enzyme for low fidelity PCRPCR purification kit (Qiagen, Hilden, Germany, Cat. no.: 28104)1 kb DNA ladder (New England Biolabs, Ipswich, MA, USA, Cat. no.: N3232)Glycerol (VWR, Radnor, PA, Cat. no.:0854), 80% *v/v* in filtered water. Autoclave to sterilize.Cryogenic tubes, 1.8 mL (Thermo Fisher Scientific, Waltham, MA, USA, Cat. no.: 377267)

### 2.2. Equipment

The equipment used in this protocol includes:ThermocyclerStandard horizontal agarose gel electrophoresis tank, gel tray, comb and power supplyAutoclave for sterilizing media, culture tubes, etc.37 °C incubator for bacterial growth37 °C shaking incubator for bacterial growth30 °C incubator with a tube rotator for yeast growthWater bath or heat block set to 42 °CMulti-tube vortex (Scientific Industries, Bohemia, NY, USA, Cat. no.: SI-D238), kept at 4 °C in cold room or refrigeratorNanodrop SpectrophotometerSpectrophotometer to measure OD_600_ of culturesMicrowave for preparing agarose gelBenchtop microfugeAccess to a DNA sequencing facilityAccess to oligonucleotide synthesis for preparation of gRNA oligos, gRNA sequencing primer, HDR template DNA, and for PCR/sequencing of target gene

### 2.3. Software

The software used in this protocol includes:Benchling (https://benchling.com (accessed on 25 September 2022)), online resource for gRNA and HDR template design. Alternatively, another gRNA design software such as E-CRISP [25]

## 3. Procedure

### 3.1. Designing the gRNA Oligonucleotides

After identifying the target gene and desired mutation, design a gRNA that is as close to the mutation site as possible. gRNA design programs, such as the one integrated into the Benchling online lab notebook service, can identify a variety of potential gRNA sites in a genomic region and can provide scores based on predicted on- and off-target activity. The Cas9 from *Streptococcus pyogenes* encoded by pCASB recognizes the protospacer adjacent motif (PAM) 5′ NGG 3′. This sequence must be present immediately downstream of the 20-bp gRNA target site [12].

Ideally, the gRNA should be contained within the sequence of the HDR template DNA that will be co-transformed along with the plasmid to create the desired mutation. This way, the template DNA can contain a silent mutation of the gRNA’s associated PAM site, preventing repeated cutting by Cas9 after the mutagenesis has taken place. Therefore, the gRNA should typically be no farther than 40–70 bp away from the desired mutation site.

To use the Benchling gRNA design tool, use the following steps (further information can be found at https://help.benchling.com/en/articles/670980-design-guide-rnas-grnas (accessed on 25 September 2022)).

Import the sequence of the gene to be edited as a DNA sequence. Search for the gene using the systematic open reading frame nomenclature (i.e., *YDR007W* rather than *TRP1*) to facilitate gene import.Select “Design and analyze guides” tool from the CRISPR menu.Set the parameters to design a 20-nucleotide guide, using the PAM sequence NGG, using the *S. cerevisiae* genome (used to calculate off-target scores).Select a region of the gene immediately adjacent to the desired mutation site (±50–100 bp is suggested).Identify the gRNA sequences which are considered good based on the on- and off-target scores: those with an on-target score [26] of 60 or higher, and an off-target score [27] of 50 or higher.Select a gRNA sequence as close to the target mutation as possible (within 70 bp of the target mutation, as discussed earlier).Make note of the chosen gRNA sequence and proceed to gRNA oligonucleotide design.

The gRNA-encoding oligos must contain the following sequences, to match the BsaI overhangs in the pCASB vector.

Top strand: 5′ CTTT 3′, followed by the 20 bp gRNA sequence

Bottom strand: 5′ AAAC 3′, followed by reverse complement of the gRNA sequence

Once the gRNA sequence has been designed, design two complementary oligos that can be cloned into the pCASB vector after digestion with BsaI. Digestion will leave behind the gRNA site shown in Figure 1B, with two 4 bp overhangs to be used for ligation of the new gRNA. For example, to generate the gRNA against the DNA sequence ATGTCTGTTATTAATTTCAC, which targets the *N*-terminus of the yeast *TRP1* gene [20], requires the following oligonucleotides:

Top strand oligonucleotide: CTTTATGTCTGTTATTAATTTCAC

Bottom strand oligonucleotide: AAACGTGAAATTAATAACAGACAT

After annealing, the paired oligonucleotides will form the double-stranded sequence shown in Figure 2, where the grey highlighted region encodes the gRNA, and the un-highlighted overhangs are available to ligate into the pCASB vector after digestion with BsaI.

### 3.2. Clone the gRNA-Encoding Annealed Oligonucleotides into the pCASB Plasmid

#### 3.2.1. Anneal the Complementary Oligonucleotides

8.Resuspend each oligonucleotide in filtered water to 100 mM.9.In a PCR tube, mix 2.5 µL of each 100 mM oligonucleotide with 20 µL of oligonucleotide annealing buffer.10.Using a thermocycler, heat oligonucleotides to 95 °C for 2 min then cool by 0.1 °C per second to 20 °C.11.Use annealed oligonucleotides within a few hours or freeze at −20 °C for future use. Their final concentration is approximately 160 ng/µL.

#### 3.2.2. Clone Annealed Oligonucleotides into pCASB Plasmid

12.Dilute annealed oligonucleotides 1:1000 in filtered water.13.Assemble a 20 µL cloning reaction, adding reagents in the following order:
Filtered water (to 20 µL final volume)3 µL of annealed oligonucleotides (~480 pg)75 ng of pCASB plasmid2 µL of T4 DNA Ligase buffer1 µL of T4 DNA Ligase enzyme1 µL of BsaIHF-v2 enzyme14.In a thermocycler, incubate reaction for 30 min at 37 °C, followed by 5 min at 60 °C. (While 30 min at 37 °C was chosen for the Golden Gate cloning (Section 3.2.2 step 3), as few as 10 min at 37 °C can lead to a successful reaction in a single-insert reaction. For troubleshooting an unsuccessful cloning, the time can be increased to 60 min)

#### 3.2.3. Transform Golden Gate Cloning Reaction into *E. coli* and Confirm Successful Cloning

15.Transform 4 µL of ligation reaction into competent *E. coli* according to manufacturer protocol.16.Grow all transformed cells overnight on LB + Kanamycin plates.17.The next day, pick 6–10 colonies using sterile wooden sticks (or pipette tips) into 2 mL LB + Kanamycin (0.4 mg/mL diluted from 1000× stock) liquid cultures in 14 mL sterile plastic tubes.18.Grow cultures overnight at 37 °C with shaking.19.Isolate plasmid DNA from each culture using a Miniprep kit or similar method.20.Measure concentration of each plasmid using a NanoDrop spectrophotometer, and send 800 ng (or as directed by the sequencing facility) of pCASB-based vector for Sanger sequencing of the gRNA site using the following sequencing primer [20]:

5′-CGGAATAGGAACTTCAAAGCG-3′

21.After obtaining sequencing results, save the samples containing successfully cloned plasmids. This DNA will be directly used for yeast transformation (step 41-i) and can be stored for many months at −20 °C.

### 3.3. Design and Amplify HDR Template DNA

#### 3.3.1. Design HDR Template

Using any preferred software, design the desired mutant locus in the target gene. Designed mutations should include a silent mutation in the PAM sequence of the gRNA (or another mutation that disrupts gRNA binding) to prevent repeated cutting by Cas9 after recombination has taken place. For most gene edits such as deletions, short insertions, or point mutations in a small region, the HDR template DNA will be designed as three 60 bp oligonucleotides that are PCR-amplified to create a 160 bp template sequence. The resulting DNA can be concentrated and transformed into yeast along with the pCASB plasmid encoding your specific gRNA. 

The three oligonucleotides are designed in a variety of ways, a few examples of which are shown in Figure 3. For most mutations, design a “middle” (M) oligonucleotide that includes all desired genetic changes, and two oligonucleotides (“forward”, upstream on the same DNA strand as the “middle”, and “reverse”, downstream and on the opposite DNA strand as the “middle”) act as primers for PCR amplification with a high-fidelity PCR enzyme (Figure 3A,B). All three oligonucleotides are 60 bp long, a common limit of standard commercial DNA oligonucleotide synthesis, and the “forward” and “reverse” oligonucleotides each overlap the sequence of the “middle” by 10 bp. The final 160 bp template has up to 70–80 bp of homology with the target gene on each side of the mutations.

Following the example of gene deletion methods such as by Longtine et al. [6] we prefer to design a template with at least 40 bp of homology to the native target gene on either side of the desired mutations. Therefore, in the case of inserting mutations that span longer regions of the target gene (i.e., multiple point mutations >80 bp apart, large insertions such as an epitope or GFP tag, etc.), the “middle” template oligonucleotide can be lengthened or replaced in multiple ways. For example, a longer oligonucleotide up to 90 bp can often be synthesized by commercial services, long enough to accommodate insertions such as a 3 × FLAG tag. Alternatively, an existing DNA template can be used for the PCR amplification, such as a plasmid or other gene containing a desired epitope tag (Figure 3C,D).

This method has been used successfully to replace over 300 bp of a gene with an alternative sequence and to add 1 kb to a gene in our hands. Efficiency of all the forms of genetic editing shown in Figure 3 is approximately equal, yielding about 90% success rate for producing a mutant strain and 50–80% positivity rate for each transformant sequenced. However, there can be some variability based mostly on the effectiveness of the gRNA and amount of template DNA provided.

#### 3.3.2. PCR Amplify the HDR Template (The following Is an Example Protocol Using Three 60–90-bp Oligos, Which Can Be Replaced with Any High-Fidelity PCR Reaction to Obtain the Desired Template)

22.Dilute the three oligonucleotides to 10 µM in filtered water23.Set up PCR reaction (we have found that two 50 µL reactions is usually sufficient for one yeast transformation, but the efficiency of the PCR can vary for each set of oligos such that the number of PCR reactions can be increased as needed).
76 µL of filtered water10 µL of Expand HF buffer2 µL of 10 mM dNTPs1 µL of “middle” template oligonucleotide5 µL of “forward” primer oligonucleotide5 µL of “reverse” primer oligonucleotide1 µL of Expand HF enzyme24.Split the reaction into two PCR tubes, 50 µL per tube.25.Perform PCR with the following parameters:
Initial denaturation: 2 min, 95 °C20 * cycles:
○Denaturation: 30 s, 95 °C○Annealing: 30 s, 55 °C * ○Elongation: 1 min *, 72 °CFinal elongation: 7 min, 72 °CParameters marked with an asterisk (*) (annealing temperature, elongation time, and number of cycles) are recommended for all PCR enzymes. Other parameters may vary.26.Ethanol precipitate PCR product (recommended for short templates, longer templates can be concentrated using a PCR purification kit (such as Qiagen):
Combine all PCR reactions into one 1.7 mL PCR tube.Add 3× volume of 100% ethanol (300 µL for two PCR reactions).Add 1/10th volume of 3 M Sodium Acetate (10 µL for two PCR reactions).**OPTIONAL STEP**: Incubate at −20 °C for 30 min or more (freezing DNA facilitates precipitation from ethanol but is not required).Centrifuge at maximum speed in a room temperature benchtop microfuge for 10 min, aspirate and discard supernatant while being careful not to disturb the clear/white DNA pellet.Wash pellet with 1 mL of 70% ethanol.Centrifuge at maximum speed in a room temperature benchtop microfuge for 5 min, aspirate supernatant.Repeat steps vi and vii, for two washes in total.Let pellet air dry for 15 min or until no ethanol remains in the tube.Resuspend pellet in 12 µL of filtered water: 10 µL for transformation reaction (below), plus 2 µL to run on an agarose gel.27.Dilute 2 µL of resuspended PCR product in 8 µL of filtered water and 2 µL of 6× DNA loading dye.28.Confirm that the PCR product is the correct size by visualizing on a 1% agarose/TAE gel containing 1× SYBR Safe DNA stain (or equivalent). Include 6 µL of 100 bp ladder).29.Image the gel on a UV transilluminator. If the PCR product is brighter than the equivalently sized band of the 100-bp ladder, there is an adequate amount of PCR product. If the PCR band is noticeably dimmer, repeat PCR with more reactions. If the band is very bright (>10× the intensity of the ladder), consider diluting the remaining DNA with additional water or using less for each yeast transformation.


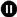
**PAUSE STEP:** the HDR template DNA can be stored for many weeks at −20 °C.

### 3.4. Transform Plasmid and HDR Template DNA into Desired Yeast Strain

30.The night before the transformation, use a wooden stick to inoculate 2–5 mL of liquid YPD media (It is not necessary to use YPD media. Other media, such as SC with amino acid dropout selections, can be used if necessary. Take care to properly prepare the media when G418 is necessary for plasmid selection. For example, ammonium sulfate present in SC media should be replaced with 1 mg/mL of monosodium glutamate) in a glass culture tube with a colony of the desired yeast strain. Grow overnight at 30 °C on a rotator.31.The next morning, measure the OD_600_ on a spectrophotometer and dilute the culture to an OD_600_ of 0.2 in at least 5 mL of YPD. Grow diluted culture at 30 °C on a rotator.32.Allow the culture to go through at least 2–3 more cell divisions, until it reaches mid-log phase (OD_600_ 0.8–1.0).33.In two 1.7 mL microcentrifuge tubes (one tube for the transformation, and one as a negative control), harvest 2 OD (~3 × 10^7^) cells.34.Centrifuge the tubes for 1 min at 13,000 RPM in a benchtop microcentrifuge at room temperature. Aspirate off and discard the supernatant.35.Resuspend each pellet in 1 mL of yeast transformation buffer.36.Vortex tubes at max speed for 30 s.37.Centrifuge the tubes for 1 min at 13,000 RPM in a benchtop microcentrifuge at room temperature. Aspirate off and discard the supernatant.38.Repeat steps 24–27.39.Resuspend cell pellet in 100 µL yeast transformation buffer.40.To each tube, add 10 µL of single-stranded DNA from salmon testes.41.Add nothing else to the negative control tube. To the other add:
100–500 ng of cloned plasmid encoding Cas9 and correct gRNA (1 µL of miniprepped plasmid DNA is sufficient).5–10 µL of prepared HDR template DNA PCR product, depending on brightness of visualized band.42.Incubate tubes on the benchtop for 10 min.43.Add 270 µL of 40% PEG solution to each tube, pipette to mix.44.Place tubes in 30 °C incubator for 30 min (no shaking or rotating necessary).45.Add 50 µL DMSO to each tube.46.Heat shock cells for 15 min at 42 °C, in either a heat block or water bath.47.Centrifuge cells for 1 min at 13,000 RPM in a benchtop microfuge, aspirate off and discard the supernatant.48.Wash pellets once with 1 mL of sterile filtered water, centrifuge at 13,000 RPM in a benchtop microfuge. Aspirate off and discard the supernatant.49.Resuspend cell pellets in 250 µL of YPD media.50.Allow cells to recover for 2 h at 30 °C with rotating.51.Spread each 250 µL culture on a YPD plate containing 400 µg/mL G418, allow to grow for 2–3 days at 30 °C for colonies to form. After this time, confirm that the negative control plate has no colonies and discard it.

### 3.5. Isolate New Strain and Confirm Gene Editing Success

#### 3.5.1. Select for Cells That Have Lost the Cas9-Expression Plasmid to Allow for Future Re-Use of the Marker

The CRISPR-Cas9 gene editing will take place over the initial 2–3 days where cells are grown on YPD + G418 to select for presence of the pCASB-based plasmid (the KanR marker confers resistance to the drug G418 in yeast). After that point, the presence of highly expressed Cas9 is not desirable. Most yeast will naturally lose the plasmid if grown without drug selection, preventing unwanted lingering Cas9 activity and freeing the KanR selection marker to be used for future genetic manipulations of the same strain.

52.Using a sterile wood stick, pick 12 individual colonies (isolates) from the transformation plate and re-streak (for single colonies) onto YPD plates.53.Grow plates for 2 days at 30 °C until single colonies form.54.From the YPD plates, pick one colony from each of the 12 isolates, and re-streak again onto:
YPD plates, for single coloniesG418 plates (all isolates can be plated as patches on a single plate, as individual colonies are not necessary) to check for plasmid loss. Yeast that has lost the pCASB plasmid should now *not* grow on G418-containing media.55.Grow plates for 2 days at 30 °C.56.Identify isolates which did *not* grow on G418.57.From the corresponding isolate on the YPD plate, use a wood stick to choose a single colony. Use the same stick to both inoculate a fresh YPD plate and a glass culture tube containing 2 mL of YPD liquid media. Streak each isolate on the fresh YPD plates for single colonies.58.Grow plates for 2 days at 30 °C and liquid cultures overnight at 30 °C on a rotator.

#### 3.5.2. Isolate Genomic DNA from Transformants (Can Be Done by Any Method, Such as a Genomic DNA Isolation Kit. Below Are Instructions for a Phenol-Chloroform Extraction)

59.Harvest 1–1.5 mL of a saturated overnight yeast culture in a 1.7 mL microcentrifuge tube by centrifugation 1 min at 13,000 RPM in a benchtop microfuge.60.Discard supernatant, resuspend cell pellet in 1 mL of filtered water.61.Centrifuge cells 1 min at 13,000 RPM in a benchtop microfuge, aspirate off and discard supernatant.62.To each tube, add 200 µL of the following:
1× TE bufferYeast genomic prep buffer0.5 mm zirconia/silica beadsPhenol:chloroform:isoamyl alcohol63.Vortex for 5 min at 4 °C.64.Centrifuge 10 min at 13,000 RPM in a benchtop microfuge.65.Tubes will have visible layers; transfer the top layer (should be 250–300 µL) to a fresh 1.7 mL tube. Discard other layers.66.Precipitate DNA from the top layer by adding 750 µL (~3× volume) of 100% ethanol and 25 µL (1/10th volume) of 3M sodium acetate. Mix tubes by inverting several times.67.**OPTIONAL STEP:** Freeze tubes at −20 °C for 30 min to facilitate precipitation.68.Centrifuge cells at 13,000 RPM in a benchtop microfuge to pellet DNA.69.Wash DNA pellets as described in steps vi–ix of step 15.70.Resuspend air-dried pellets in desired amount of water (~300 µL will yield a DNA concentration that can be used directly as a template in the subsequent PCR reaction for screening).

#### 3.5.3. Confirm Gene Editing Success by PCR and Sequencing

71.Design PCR primers flanking the region of interest, to amplify a region of at least 250 bp around the site of the mutation.72.Design a sequencing primer at least 100 bp upstream of the desired mutation (Sanger sequencing will begin >50 bp downstream of the sequencing primer).73.Using any standard PCR polymerase enzyme (high-fidelity not necessary), amplify the desired site.74.**OPTIONAL STEP:** Run a small portion of each PCR product (~4 µL of a 25 µL reaction) on a 1% agarose/TAE gel with 1 kb ladder to visualize a product of the correct size).75.Use a PCR purification kit to isolate the PCR products76.Send PCR product from each isolate for Sanger sequencing to confirm gene editing success.77.After obtaining sequencing results and identifying successful mutants, the yeast strains are complete. If desired, make glycerol stocks and store at −80 °C (Glycerol stocks should be made of multiple isolates due to the chance of off-target effects. While off-target mutations as a result of CRISPR-Cas9 in yeast have been found to be rare [14], we recommend performing initial assays on three (or more, if desired) isolates to screen for uniform phenotypes):
Using a wood stick, pick a single colony of each desired isolate and inoculate 2 mL of YPD liquid media in a glass culture tube.Grow overnight at 30 °C on a rotator.The next day, in a cryogenic tube, combine 800 μL of the saturated overnight culture with 200 μL of 80% glycerol. Store at −80 °C.

## 4. Expected Results and Troubleshooting

The *E. coli* transformation (Section 3.2.3) should yield numerous individual colonies. In the event there are no colonies, or sequencing of the plasmid shows that the oligonucleotides were not ligated in, consider a longer incubation of the Golden Gate cloning (While 30 min at 37 °C was chosen for the Golden Gate cloning (Section 3.2.2 step 3), as few as 10 min at 37 °C can lead to a successful reaction in a single-insert reaction. For troubleshooting an unsuccessful cloning, the time can be increased to 60 min).

Yeast transformation (Section 3.4) should yield many yeast colonies on YPD plate containing G418. If fewer than twelve yeast colonies are present on the transformation plate, isolates can still be screened, but the possibility of inaccurate editing might be higher. Low colony numbers may indicate high gRNA activity without adequate HDR, as repeated cutting of the same site by Cas9 can be toxic to cells [23]. The most likely explanation for low numbers (or absence) of yeast colonies after transformation is a too low amount of HDR template DNA. Try repeating the transformation with DNA from a larger number of PCR reactions (to yield a higher DNA concentration) or re-designing the template to obtain higher PCR efficiency.

We recommend picking twelve yeast colonies to yield a successfully genetically edited yeast strain. When editing is successful, efficiency is greater than 50%; sequencing twelve colonies yielded sufficient isolates to save at least three for further experiments. Similar to what was reported by Ryan et al. [14], mutation efficiency was occasionally near 0% for approximately 5–10% of transformations. This could be remedied by the selection of a different gRNA to the same genomic region. When using this method, if after sequencing none of the selected colonies contain the desired mutation, it is possible but unlikely that picking more colonies may yield the desired result. Consider repeating the transformation one time with an increased amount of HDR template DNA, but we also recommend selecting at least one or two new gRNA sequences in the same target region. Other ways to speed up the identification of genetically engineered yeast if a lot of screening is required is to perform single colony PCR, followed by sequencing. It may also be pertinent to design an HDR template that includes a base change(s) that generate a new restriction enzyme site or inactivates an existing site, to enable screening of the PCR products by restriction enzyme digestion.

## Figures and Tables

**Figure 1 mps-05-00079-f001:**
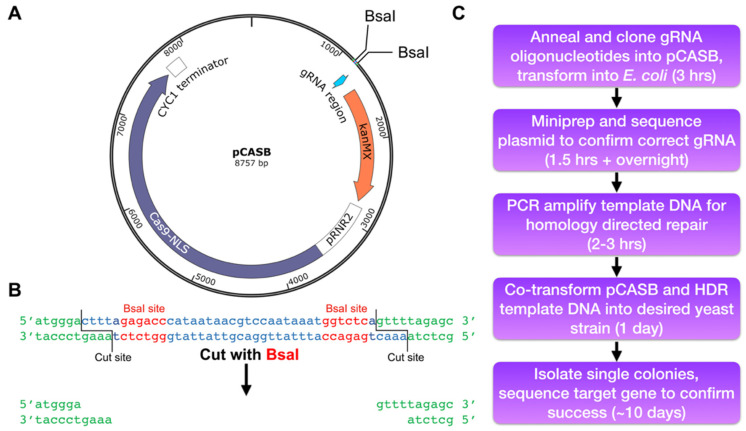
(**A**) Schematic of the pCASB plasmid, highlighting the locations of the BsaI cut sites. (**B**) The modified gRNA locus of pCASB. BsaI recognition sites are labeled in red, DNA lost after cutting is labeled blue, and unaltered plasmid backbone left behind after cutting in green. (**C**) Schematic of the CRISPR-Cas9 gene editing protocol, starting with gRNA cloning and ending with a modified yeast strain.

**Figure 2 mps-05-00079-f002:**
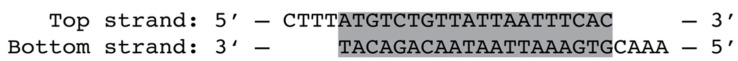
Example of the double-stranded DNA sequence and 4 bp overhangs resulting from annealing of gRNA oligonucleotides, before ligation into the pCASB vector. The sequence highlighted in grey encodes the gRNA 5’- ATGTCTGTTATTAATTTCAC -3’, targeting the N-terminus of the yeast *TRP1* gene.

**Figure 3 mps-05-00079-f003:**
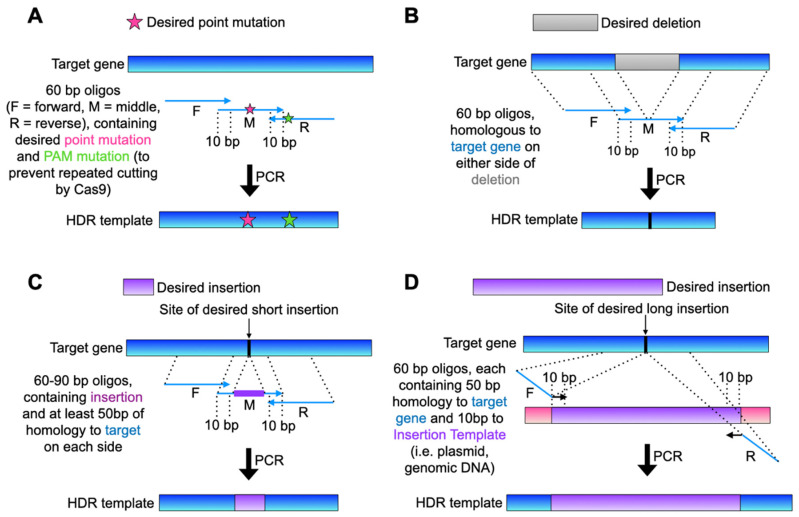
Representation of the HDR template design for a (**A**) few base-pair point mutations, (**B**) large deletion, (**C**) short (≤80 bp) insertion, or (**D**) long (>80 bp) insertion into the yeast genome. Note: as indicated in Section 3.1, a silent mutation of the PAM (or another other portion of the gRNA target sequence) should be always included if the mutation does not otherwise disrupt the gRNA target, to prevent repeated cutting by Cas9 after repair is complete.

## Data Availability

Not applicable.

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
