# Peer review of "A Simple, Improved Method for Scarless Genome Editing of Budding Yeast Using CRISPR-Cas9"

_mps, 2022, doi:10.3390/mps5050079_

Round 1

Reviewer 1 Report

Aguilar et al is describing in the “A simple, improved method for targeting genome editing of budding yeast using CRISPR-Cas9” a methodology to genome edited the yeast genome. In this protocol, the author is using a cloning strategy that relies on having the Cas9 and the sgRNA guide in one plasmid. As the author states, the described protocol builds on an already published one by Ryan et al, to which it makes only an incremental change: insertion of BsaI restriction sites flanking the gRNA insertion site so that the gRNA’s sequence can be exchanged with another. Based on the introduction that the author is providing and the current state of gene editing, I fail to understand the advantage of this protocol and the novelty of it.

I found the introduction (lines 28-36) to be very confusing. It should clearly state the rationale for developing efficient ways to genome edit the yeast genome. However, this first paragraph, as written, is very confusing. It states that the HR machinery is highly active in the yeast genome, an important factor for gene editing. It then states that because HR machinery is highly active a DNA template with homology arms can be easily incorporated into the genome without having to introduce double strand breaks. Then why is the author developing this protocol if a CRISPR/Cas9 is not necessary in yeast?

On line 33, the author states that “a selectable marker was required on the template DNA for selection of the infrequent genomic insertion events”. This is a direct contradiction from previous statement.

Lastly, on line 34 is states that the selectable marker needs to be removed. This marker can be clones in a way that doesn’t interfere with proper gene expression, and can be a florescence maker, rather than a metabolic marker, eliminating the need to remove it.

Current CRIPR/Cas9 gene editing strategies are relying on Cas9 protein and sgRNA guide (mRNA chemically modified) that are nucleofected as a complex into cells. Why is the author not exploring this latest approach which bypasses the need of cloning? The DNA template could still be nucleofected as a plasmid and it can be designed to carry a selectable marker to follow the modified cells throughout the experiment.

What the author is proposing seems too laborious and unnecessary in light of so much advancement in the field of gene editing.

Reviewer 2 Report

The authors have described a detailed and improved method for CRISPR-Cas9 genome editing protocol to introduce multiple types of directed genetic changes into the yeast genome. They have done significant amount of work in detailing the material methods of the total protocol. They also have pointed out the different kind of modification (point mutation and insertion/ deletion) that can be introduced using this method.

But one major drawback is they haven’t mentioned how efficiently this different type of mutations can be introduced. They did mention an efficiency of 0-100% which is kind of confusing. I would strongly recommend the authors to provide more concreate mutation specific editing efficiency information.

Line 251-252: Software tools used to design the guide RNA and the parameters should be described clearly.

Line 310-319: Colony PCR followed by sanger sequencing can be used to check the correct cloning and incorporation of gRNA at this stage. Why did they go for plasmid extraction?

Line 429: Concentration of salmon testes DNA is not mentioned.

Line 484: Composition of Yeast genomic prep buffer should be mentioned

Reviewer 3 Report

Aguilar et al., have submitted a methods article describing targeted genome editing using CRISPR-Cas9 tools in budding yeast. Overall, the manuscript is very well written, and detailed and will be an important tool to the community of yeast biology. The expected results and troubleshooting tips are important sections that the authors have very nicely explained. However, I have the following suggestions for the authors to consider.

1.     Referring to figure 2, Please mention the efficiency of point mutation, insertion, and deletion using this protocol.

2.     What strategies are recommended to screen for off-target effects?

Round 2

Reviewer 1 Report

Dear Authors,

Thank you for addressing my comments. I believe that a concise version of your answer to point 4 (as to why no other current version is amenable to genome editing the yeast genome) should be included in your introduction.  This way, the reader will understand that your protocol represents the most up-to-date method for yeast research, unlike mammalian cells, where more progress has been made.

All the additional textual modifications that were made to the introduction and remaining of the protocol are acceptable.

I find the protocol and schematics easy to follow.

Author Response

Response to Reviewer 1

Point 1: "I believe that a concise version of your answer to point 4 (as to why no other current version is amenable to genome editing the yeast genome) should be included in your introduction."

Response 1: We thank the reviewer for their feedback on the introduction. We have added this requested information, clarifying that plasmid and DNA transformation-based methods are the most up-to-date and facile technique in the yeast CRISPR-Cas9 field.